# Experimental Evaluation of Foam Diversion for EOR in Heterogeneous Carbonate Rocks

**Motaz Taha [1], Pramod Patil [2] and Quoc Nguyen [1,\*]**

[1] Hildebrand Department of Petroleum and Geosystems Engineering, University of Texas at Austin, Austin, TX 78712, USA

[2] Rock Oil Consulting Group, Sugar Land, TX 77479, USA

\* Correspondence: quoc_p_nguyen@mail.utexas.edu; Tel.: +1-512-471-1204

**Abstract:** Immiscible gas injection applied to heterogeneous carbonate reservoirs can be inefficient due to poor conformance control. Foam mobility control is proposed in this work as a solution for gas conformance issues in such reservoirs. A unique experimental program was developed to evaluate alkyl polyglucoside (APG) stabilized foam for foaming ability, emulsion-forming tendency and resistance to oil. Dynamic methane foam behavior is systematically studied through single and dual injection core flooding experiments, simulating foam diversion during immiscible methane flooding in a layered reservoir with a significant layer permeability contrast. Results show a stable foam-oil system with no viscous emulsions at very high formation brine salinity (144,000 ppm total dissolved solids). Single-core floods for the high permeability layer (Unit-A) showed that foam viscosity of 27 cP could be achieved at 11% oil saturation ($S_o$). Under similar oil-wet condition, the low permeability zone (Unit-B) could generate foam of 21 cP at 18.9% $S_o$, indicating an increase in injected fluid mobility reduction with permeability. Dual-core injection experiments, which is designed to evaluate accurately fluid diversion capacity of such foams, reveals remarkable dynamic foam behaviors. While the water-wet condition indicates the scalability of foam behaviors (i.e., the ability of foam to control fluid mobility against the variation of rock permeability) between the single and composite core systems, the oil-wet condition confirms good foam resistance to residual oil that resulted in an increase in Unit B production from 46 to 82%, and 74 to 85% for Unit-A. Moreover, dual-core floods representing premature waterfloods (i.e., higher oil saturation) shows even more dramatic incremental oil recovery (44 to 81% in Unit-A and 17.5 to 71% in Unit-B), evidencing the ability of foam to self-viscosify with permeability variation at varying oil saturations.

**Keywords:** foam; conformance control; water-alternating-gas (WAG) injection; enhanced oil recovery; oil-wet carbonate

## 1. Introduction

The field studied in this work is made up of cretaceous carbonate formations with an overlaying sandstone formation. The field is developed with horizontal wells placed in radial or parallel drive patterns of alternating injector and producer wells. The reservoir is composed of thin carbonate layers of contrasting permeabilities that can be categorized into two main groups. First, Unit-A is 60–100 mD, and Unit-B layers are 10–20 mD. The field has undergone waterflooding, and then water-alternating-gas injection (WAG) with hydrocarbon gas that is immiscible at the reservoir conditions of 1400 psi and 55 °C. Produced gas-oil-ratio (GOR) and oil properties (density and viscosity) vary in different parts of the reservoir.

Typically, the incremental recovery achieved with WAG is less than predicted [1,2]. Injected gas is lighter than oil, and less viscous, which causes it to finger through the oil by viscous fingering [3]. Other conformance issues occur in cases where gas can rise to the top of the structure unevenly due to gravity override, or channel through high permeability

streaks [4,5]. The above conformance issues, as well as the variation in oil density and GOR along different layers of the reservoir explains the low efficiency of the implemented WAG process in this reservoir. The oil-wet nature of the carbonate reservoir may also contribute to the conformance issues during WAG injection because water relative permeability increases with the degree of oil wetness, which further increases the total mobility of injected gas and water as the non-wetting phases, giving rise to fluid fingering or channeling [6,7].

Enhanced oil recovery (EOR) processes require mobility control to maximize sweep efficiency [8]. Foam, a dispersion of a gas and an aqueous phase stabilized by specifically tailored surfactants has been reported to achieve mobility control and thus enhancing the overall oil recovery efficiency [9–12].

Foam injection results in the dispersed flow of two phases where the combined mobility is less than either individual phase mobility, thus resulting in improved mobility control and oil displacement [13,14]. In addition, foam apparent viscosity increases with permeability [15–17]. Subsequently, the dispersed-gas mobility decreases with increasing rock permeability. Foam generation by snap-off and lamella division requires gas to invade into the dominant pores and aqueous foam films (or lamellae) to be mobilized, which occur when a critical capillary number or a critical pressure gradient is exceeded. These critical conditions for foam generation have been proved experimentally and theoretically to be directly correlated with the medium permeability [18–21].

This cannot be achieved with conventional polymers for chemical EOR [22,23]. Foam injection as foam assisted WAG (FAWAG) or surfactant alternating gas (SAG), provides a low-risk mobility control solution that is reversible and tunable through surfactant properties, injection rate, as well as injected gas fraction or foam quality (FQ), defined as the ratio of injection gas rate to the total injection rate of gas and water. An increase in foam quality makes foam drier and less stable due to the resulting increase in local capillary pressure [18]. Furthermore, studies show that applying foam technology in the field has an overall reduced carbon footprint when compared to normal WAG, or long-held practices such as flaring uneconomical field gas [24].

Many field pilots can be found in the literature. Recent examples include a $CO_2$ foam pilot conducted in the East Seminole Field, in the Permian Basin. Results of this trial show clear evidence of foam generation (elevated SAG cycle pressures), and a 30% improvement in oil recovery over baseline projections [25]. Another example is the project implemented in Salt Creek Field in Wyoming, whereby foam conformance improved recovery by over 25,000 bbl of oil, and reduced gas injection requirements by over 22% [26]. A $CO_2$ pilot was successfully implemented in a heterogeneous carbonate reservoir in West Texas, whereby a clear improvement in conformance was demonstrated. This led to an improvement in the oil-cut decline [27]. The foam pilot conducted in the East Vacuum Grayburg Sand Andres unit (sandstone reservoir) further demonstrates the viability of foam for conformance control, which led to a 20–60% increase in the oil production during the pilot [28].

The key objective of this work is to quantify the improvement in oil recovery from a heterogeneous carbonate reservoir when foam injection is used. Towards this end, the reservoir layers are represented with outcrop rocks of comparable properties. Previous works [29–31] screened multiple surfactant types for high salinity tolerance (referred to as aqueous stability) and foamability. An alkyl-polyglycoside (APG) surfactant was chosen having the best tolerance to the reservoir conditions. Bulk foam tests and core floods in water-wet conditions in the absence of oil found an optimum surfactant concentration of 0.35 wt.% (3500 ppm). In this study, the APG foam is characterized in terms of bulk foam stability, emulsion tendency and IFT properties against oils present in the two reservoir units. Foam core flooding in single cores is then conducted to study foam rheology and oil recovery in individual reservoir units. Finally, dual-core injection experiments are designed to evaluate the conformance improvement and fluid diversion capacity of foam.

Dual injection experiments have been used in literature for foam-acid diversion experiments [32–35], whereby these works focused on the ability of foam to divert acid to damaged zones. These works were conducted under water-wet conditions without

the presence of oil. Dual injection experiments were also used to study the impact of reservoir heterogeneities on oil recovery during supercritical $CO_2$ flooding under miscible conditions [36,37].

In this work, the objective of dual injection experiments is twofold. The first objective is to quantify the conformance control provided by foam based on the measured pressure drop across the two cores representing two reservoir flow units and the produced liquid fractional flow from each unit defined in Equation (1).

$$X = (Q_{effluent}/Q_{injected}) \tag{1}$$

where $Q_{effluent}$ is the liquid flow rate produced from each core, and $Q_{injected}$ is the total liquid injection rate. The second objective of the dual injection experiments is to quantify the oil recovery improvements due to foam conformance control in both reservoir units.

The following sections outline the materials and methods used, followed by a detailed results and discussion of characterization experiments as well as single- and dual-core flood experiments. Finally, a summary of the key findings is presented.

## 2. Materials and Methods

The surfactant used in this work was previously screened from a group of APG surfactants [25–27]. It was chosen for its high tolerance to high salinity. It has a critical micelle concentration (CMC) of 0.006 wt.%. The aqueous stability and emulsion tendency are first evaluated under reservoir conditions (55 °C, 144,000 ppm salinity).

The surfactant used in this work is first diluted to 2.5 wt.% stock solution in deionized water, and then subsequently diluted down to the required concentration (0.1–0.5 wt.%) at the formation brine salinity 144,000 ppm total dissolved solids (TDS). The composition of formation brine for both layers is listed in Table 1.

**Table 1.** Formation brine salinity (g/L).

| Salt | Brine Salinity (g/L) | Ionic Strength (mole/L) |
| :---: | :---: | :---: |
| NaCl | 99.80 | 1.708 |
| KCl | 3.86 | 0.052 |
| $SrCl_2 \cdot 6H_2O$ | 1.17 | 0.024 |
| $CaCl_2 \cdot 2H_2O$ | 33.31 | 0.599 |
| $MgCl_2 \cdot 6H_2O$ | 29.81 | 0.519 |
| Total | 144.144 | 2.901 |

The crude oil samples were filtered through 1.2 μm, 0.45 μm, and then 0.22 μm filters. The crude oil from Unit-A has a density of 0.94 g/cc (19° API at room temperature) and a viscosity of 44.3 ± 0.15 cP at the reservoir conditions. Unit-B has a different crude oil with a density of 0.84 g/cc (37° API at room temperature) and a viscosity of 10.1 ± 0.15 cP at reservoir conditions. Two outcrop rocks were used to represent the high and low permeability carbonate units of the reservoir. These rocks were sourced from Kocurek Industries. Core samples 30.48 cm long and 3.81 cm diameter were used in the foam flooding experiments. The general properties of the outcrop rocks are listed in Table 2. The specific properties of the cores taken from these outcrop samples are listed in Table 3. Methane gas at 97.7% purity provided by *PRAXAIR* was used as the injection gas for all foam flooding experiments.

**Table 2.** Properties of Unit-A and Unit-B representative outcrop rocks.

| Reservoir Layer | Outcrop | Permeability (mD) | Porosity (%) |
|---|---|---|---|
| Unit-A | Estaillades | 70–100 | 22–25 |
| Unit-B | Austin Chalk | 10–20 | 28–31 |

**Table 3.** List of core flood experiments.

| Experiment | Mode | Outcrop | Permeability (mD) | Conditions | Porosity (%) | Initial $S_{oil}$ (%) |
|---|---|---|---|---|---|---|
| Unit-A–Exp 1 | Single Injection | Estaillades | 103 | Water Wet–No Oil | 23.2 | 0 |
| Unit-A–Exp 2 | Single Injection | Estaillades | 100.8 | Oil Wet–at Sor | 25.7 | 60.9 |
| Unit-B–Exp 1 | Single Injection | Austin Chalk | 15.7 | Water Wet–No Oil | 29.5 | 0 |
| Unit-B–Exp 2 | Single Injection | Austin Chalk | 19.7 | Oil Wet–at Sor | 31.4 | 57.5 |
| DI–Exp 1 | Dual Injection | Estaillades | 50.3 | Water Wet–No Oil | 23.0 | 0 |
|  |  | Austin Chalk | 11.9 | Water Wet–No Oil | 27.5 | 0 |
| DI–Exp 2 | Dual Injection | Estaillades | 39.0 | Oil Wet–at Sor | 22.9 | 61.5 |
|  |  | Austin Chalk | 9.8 | Oil Wet–So > Sor | 26.5 | 59.0 |
| DI–Exp 3 | Dual Injection | Estaillades | 39.0 | Oil Wet–at So > Sor | 22.9 | 59.3 |
|  |  | Austin Chalk | 9.8 | Oil Wet–So > Sor | 26.5 | 57.7 |

### 2.1. Foam Stability Experiments

Bulk foam stability experiments are conducted with and without crude oil from each of the reservoir layers. A total of 10 mL of surfactant solution is placed in 30 mL glass vials and placed in a 55 °C oven. This achieves an initial foam quality of 66% in these tests. The vials are shaken for 10 s to disperse air into the surfactant solution to generate foam. The height of foam column generated is measured and recorded at regular time intervals. Foam stability represented by the foam half-life, which is defined as the time taken for the foam height to decay by half.

### 2.2. Emulsion and Aqueous Stability Tests

To assess the behavior of the APG surfactant solutions when interacting with the crude oils, emulsion phase behavior tests are carried out. First, 0.35 wt.% surfactant solutions at 45,000, 85,000, 144,000 and 180,000 ppm *TDS* are prepared. The aqueous stability of these surfactant solutions is evaluated at the reservoir temperature (55 °C) by visually observing for turbidity or phase separation over a period of two weeks. To test the emulsion tendency of these surfactant solutions, 1 mL of crude oil and 2 mL of surfactant solution are added into a glass pipet, which is sealed, mixed gently, and then monitored for phase change over 5 days at 55 °C.

### 2.3. Interfacial Tension Measurements

Interfacial tension (IFT) measurements between oil, surfactant solutions, and air are conducted using the pendant drop method. The oil-water IFT measurements are conducted at 55 °C and atmospheric pressure using a Rame-Hart Model 290 high precision tensiometer. Air is used instead of methane to represent the gas phase. Surfactant solution is placed in an optical cell. Then, an oil droplet was injected slowly using a micro-needle. A brine density of 1.05 g/cc, and air density of $1.23 \times 10^{-3}$ g/cc are used in the interfacial tension measurements. The measurements are repeated 10 times for each fluid system and the average IFT is reported with a standard deviation of 0.02 mN/m. The same method is also used to measure surfactant-air and crude oil-air *IFT* for crude oil from both reservoir layers.

## 2.4. Core Flooding Experiments

Core flooding experiments are conducted using the experimental set-up shown in Figure 1.

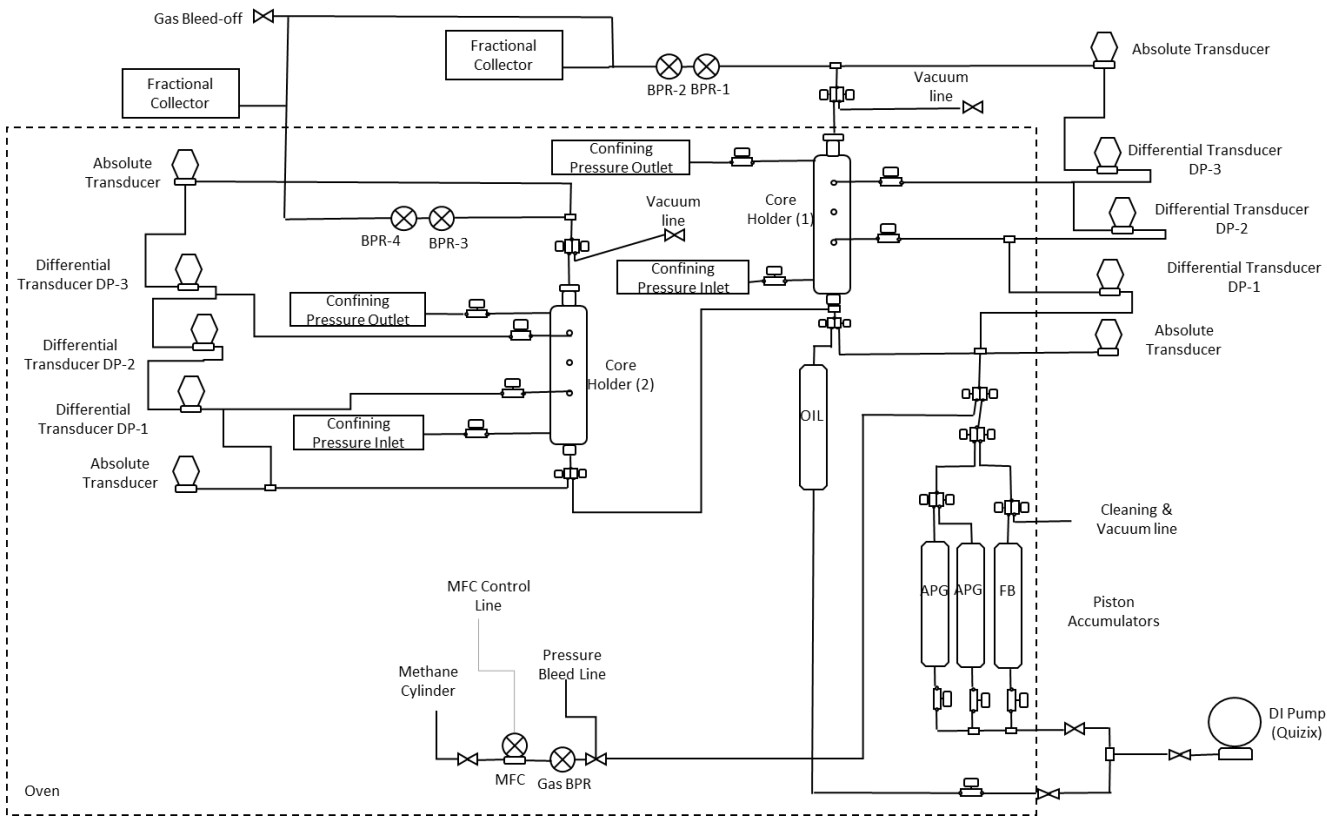

**Figure 1.** Core Flooding setup configured to single or dual injection core floods.

The core is first vacuumed for 24 h before being saturated with the formation brine. Injection is done by a Quizix pump, which displaces a piston accumulator at a controlled rate. The core porosity and permeability are then measured. Gas injection is controlled by a Matheson mass flow controller (MFC).

Foam experiments are conducted by co-injection of surfactant solution and methane gas. The effluent goes through a set of two back pressure regulators (BPRs) that ensure the core outlet pressure is maintained close to the reservoir pressure of 1450 psi. Steady state pressure drops across the core are recorded every 30 s by the differential transducers DP1, DP2, and DP3 for each core holder. The set-up is used for both single-core flood experiments as well as dual-core flood experiments, whereby a single inlet header feeds both cores, simulating the case of a layered reservoir composed of hydraulically disconnected layers.

The maximum foam viscosity is obtained under water-wet conditions in the absence of crude oil (Unit-A Exp 1 and Unit-B Exp 1) as per Table 3. In these experiments, the impact of surfactant concentration and injection rate in each reservoir layer is evaluated. Injection is conducted at 1–5 ft/d, and FQ is varied from 30 to 90%. Oil wet cores are prepared by flooding brine saturated cores with oil until the oil cut is 100%. Cores are then aged at 55 C and 1400 psi for 8 weeks. Oil wet single-core floods (Unit-A Exp 2 and Unit-B Exp 2) investigate the foam viscosity at residual oil saturation ($S_{or}$) following a water flood with formation brine until the water cut is 100%. The injection rate used in the above experiments is 2 ft/d with FQ varying from 30 to 90%.

As also shown above in Table 3, three dual-core floods are conducted. For DI-Exp 1, the cores are individually saturated with brine, and their porosity and permeability are determined. Brine is injected into both cores at 5 ft/d and the baseline fractional flow

(Equation (1)) in each core is measured. Foam injection is conducted at 5 ft/d and 50% FQ to quasi-steady state. Gas saturation, fractional flow (X), and the pressure drop across the cores are measured as foam propagates. Cores from different outcrop rocks (Estaillades, Austin Chalk) were chosen based on the contrast in porosity and permeability and the similarity in mineral composition (with more than 99 wt.% calcite and very small clay content). This contrast simulates reservoir heterogeneity and is required for clearly demonstrating the impact of foam conformance in dual injection experiments. It is noted that the wettability of natural oil-free carbonate rocks is water wet due to the high surface energy of naturally occurred calcite [38].

In DI-Exp 2, both cores are oil saturated and aged for 8 weeks. Waterflooding is conducted until the water cut from Unit-A is 100%. Foam injection is then started by co-injecting surfactant solution and gas at 5 ft/d and 50% FQ. Fractional flow (X) and oil recovery/saturation from both cores are measured. For DI-Exp 3, the cores are individually saturated with crude oil from Unit-A and Unit-B and aged again for 8 weeks. DI-Exp 3 follows the same procedure as DI-Exp 2 with the marked difference of limiting the water flood to only 1 total pore volume (PV). This is designed to start the foam flood at an oil saturation being higher than $S_{or}$ in both cores. PV in the above experiments refers to the combined pore volume from both cores.

## 3. Results and Discussion

The following sections outline the characterization experiments studying interaction of APG and crude oils from the reservoir units, followed by the results of single core flood experiments representing conditions of both units. Finally, the results of the dual injection experiments are discussed.

### 3.1. Fluid Charachterization

3.1.1. Aqueous Stability

APG solutions at 0.35 wt.% concentrations and different salinities were placed in an oven at 55 °C for a period of two weeks. Figure 2 shows the solutions at the end of the test. As shown in Figure 2, APG solutions are stable (clear) at salinities above 144,000 ppm. Some clouding occurs at the highest salinity of 180,000 ppm. Therefore, surfactant precipitation is not expected to occur at the formation brine salinity of 144,144 ppm (Table 1).

3.1.2. Emulsion Tendency

The purpose of this test is to ensure that APG and crude oil from either reservoir unit do not form unwanted high viscosity emulsions that may cause irreversible formation damage, severe surfactant retention in the reservoir, and surface separation. It is noted that the gentle mixing of oil and water in pipet is to mimic the dispersion of oil and water into each other by nature of two-phase flow in porous media, which enhances the formation of emulsions in situ. While emulsion droplet sizes generated in pipets could be different from that in porous media, the formation of undesired viscous emulsions observed in bulk emulsion tests could also occur in oil-water flow in porous media.

It is also important to ensure that pressure drops observed during core flood experiments and subsequent field tests, are a result of the in situ foam propagation and its impact on fluid mobility reduction, and not due to the formation of adverse emulsions. A snapshot of the emulsion tendency after 48 h. for each crude oil is presented in Figure 3. As shown in this figure, Winsor Type I microemulsions form with an upper oleic phase and a lower aqueous phase with solubilized oil for Unit-A oils at salinities 45,000–144,000 ppm. At the highest salinity (180,000 ppm), a middle phase of emulsions is seen in Figure 3c. However, the observed emulsions were not viscous and unlikely to form at the reservoir salinity (144,000 ppm), and thus not a concern when flowing in unit-A. It is noted that since the emulsion middle phase for the 180,000 ppm vial was very thin and had a relatively low color contrast to the access oil, it is difficult to be clearly seen in Figure 3c. However, the bottom aqueous phase can be clearly seen to have more oil-in-water emulsions than the

other vials that toke longer to separate, further indicating that this salinity causes more emulsions to form with the crude oil.

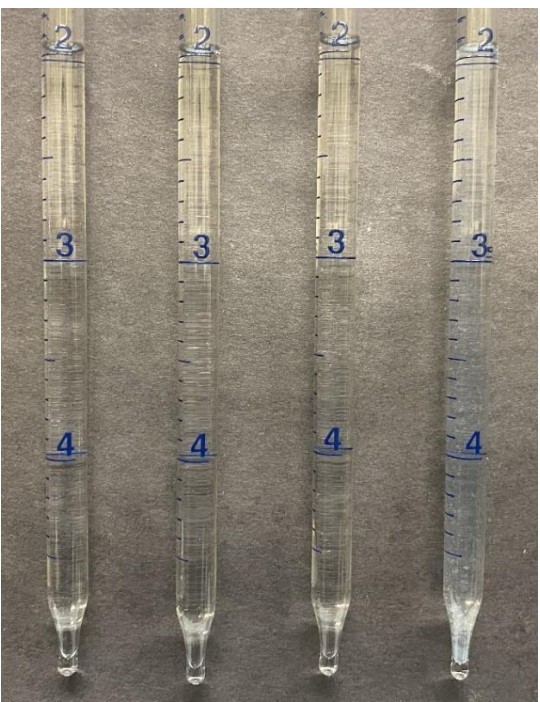

**Figure 2.** Aqueous stability tests of APG solutions at 45,000 (**Left**), 85,000, 144,000, and 180,000 ppm (**Right**). Pipette graduations shown are in mL.

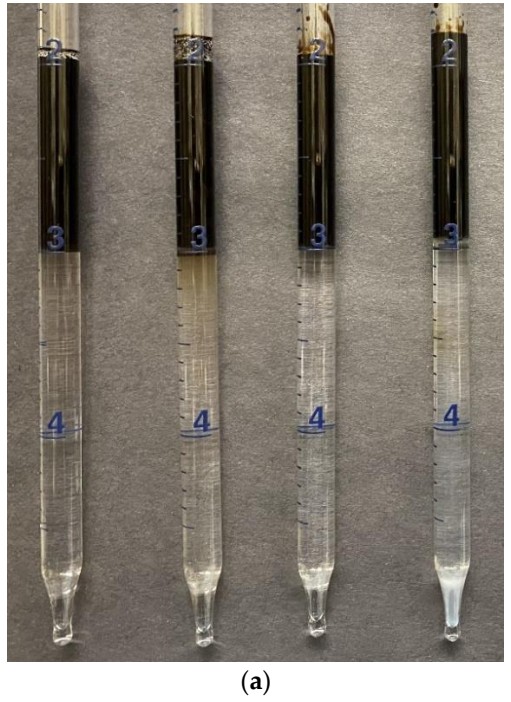
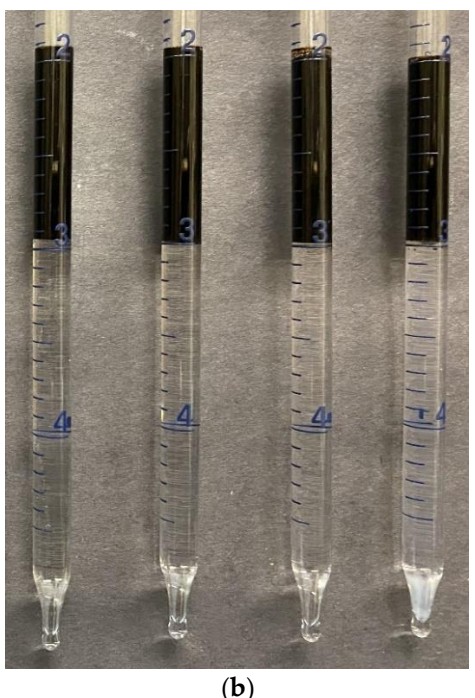

(**a**)                                                                                          (**b**)

**Figure 3.** *Cont.*

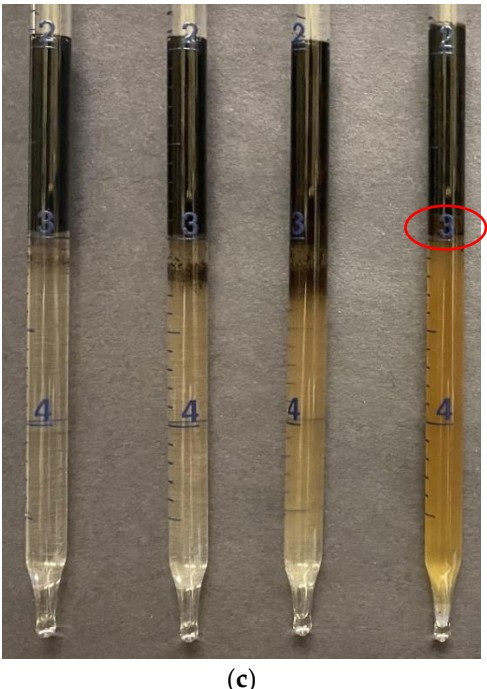
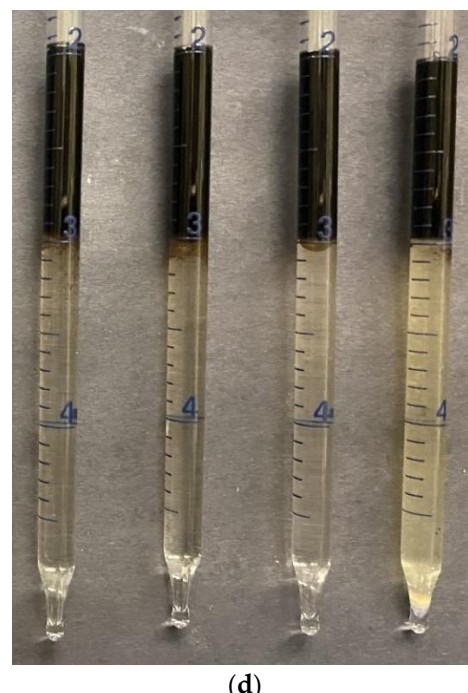

(**c**)             (**d**)

**Figure 3.** Emulsion tendency tests of APG with Unit-A and Unit-B crude oils at 45,000 (**Left**), 85,000, 144,000, and 180,000 (ppm) salinity (**Right**). (**a**) APG-Unit-A oil before mixing (**b**) APG-Unit-B oil before mixing (**c**) APG-Unit-A after mixing and resting for 2 days at 55 °C. (**d**) APG-Unit-B after mixing and resting for 2 days at 55 °C. Pipette graduations shown are in mL. The red circle in (**c**) highlights the thin layer of emulsions at 180,000 ppm.

Unit-B oil forms Winsor Type-I microemulsions at all the salinities, with less staining of the water phase (degree of oil-in-water emulsification). It can be concluded that no viscous emulsions form for this type of surfactant-oil interaction at the reservoir conditions of both Unit-A and B. Furthermore, there is no notable change in the water/oil phase volumes, indicating no significant oil solubilization. Thus, during foam injection, it is not a concern regarding emulsions as a source of flow impedance and foam can be considered as the main cause of fluid mobility reduction and conformance control.

3.1.3. Entering and Spreading Tendency of Oil on Foam Film

There are several theories quantifying foam destabilization by oil. The first of these is the defined entering (E) and spreading (S) theory. It proposes that oil droplets enter a foam film and spread along its surfaces, leading to film rupture. E and S coefficients are estimated in Equations (2) and (3) whereby E < 0 indicates the unlikelihood of oil droplets to invade foam films and spread along its surfaces [39–41].

$$E_{\text{Oil-Water}} = \sigma_{\text{(Water-Gas)}} + \sigma_{\text{(Oil-Water)}} - \sigma_{\text{(Oil-Gas)}} \tag{2}$$

$$S_{\text{(Oil-Water)}} = \sigma_{\text{(Water-Gas)}} - \sigma_{\text{(Oil-Water)}} - \sigma_{\text{(Oil-Gas)}} \tag{3}$$

where $\sigma$ is the interfacial tension between two phases.

A second theory known as the lamella number theory [42] is based on the concept that foam is destabilized by the formation and movement of emulsified oil drops that move into foam film plateau borders. Once enough droplets accumulate and join the foam films are ruptured. The lamella number is shown in Equation (4).

$$L = [r_o/r_p] \approx 0.15 \times [\sigma_{\text{(Water-Gas)}} / \sigma_{\text{(Oil-Water)}}] \tag{4}$$

where $r_o$ is the oil drop/surface radius, and $r_p$ is the foam plateau border radius, which describes the tricuspid border region between three foam lamellae. Stable (type A) foams are defined by $L < 1$, whereby less stable foam-oil systems defined as type-B have $1 < L < 7$. Very unstable type-C foams are characterized by $L > 7$. Interfacial tension measurements and the calculated E, S and L numbers for the surfactant-oil-brine systems used in this work are presented in Table 4.

**Table 4.** Interfacial tension measurements and oil-foam interaction parameters.

|  | Unit-A Crude Oil | Unit-B Crude Oil |
|---|---|---|
|  | Interfacial/surface tension [mN/m] | Interfacial/surface tension [mN/m] |
| $\sigma_{(Oil-Water)}$ | 0.39 | 0.87 |
| $\sigma_{(Oil-Gas)}$ | 45.5 | 25.1 |
| $\sigma_{(Water-Gas)}$ | 10.7 | 10.7 |
| E | −34.4 | −13.5 |
| S | −35.2 | −15.3 |
| L | 4.2 | 1.84 |

The results of Table 4 show negative E and S coefficients against both crude oils. Based on this theory, the APG stabilized foam is expected to be stable when encountering these crude oils. The L number falls in the type-B foam range, which indicates some instability due to emulsification of oil by the APG surfactant. The presence of type-1 microemulsions supports this conclusion. Furthermore, L for Unit-B crude oil is lower indicating a lower degree of emulsification as seen in Figure 3c.

### 3.1.4. Bulk Foam Stability

Bulk foam tests are conducted to investigate the APG foam stability in the presence of crude oil at the bulk foam scale. APG solution at 0.35 wt.% concentration and 144,000 ppm salinity is placed in 3 vials. The first vial does not contain any oil, and vials 2 and 3 contain 1 mL of crude oil from Unit-A and B successively. Respective snapshots of the fluid mixtures at 0, 30 and 90 min are shown in Figure 4.

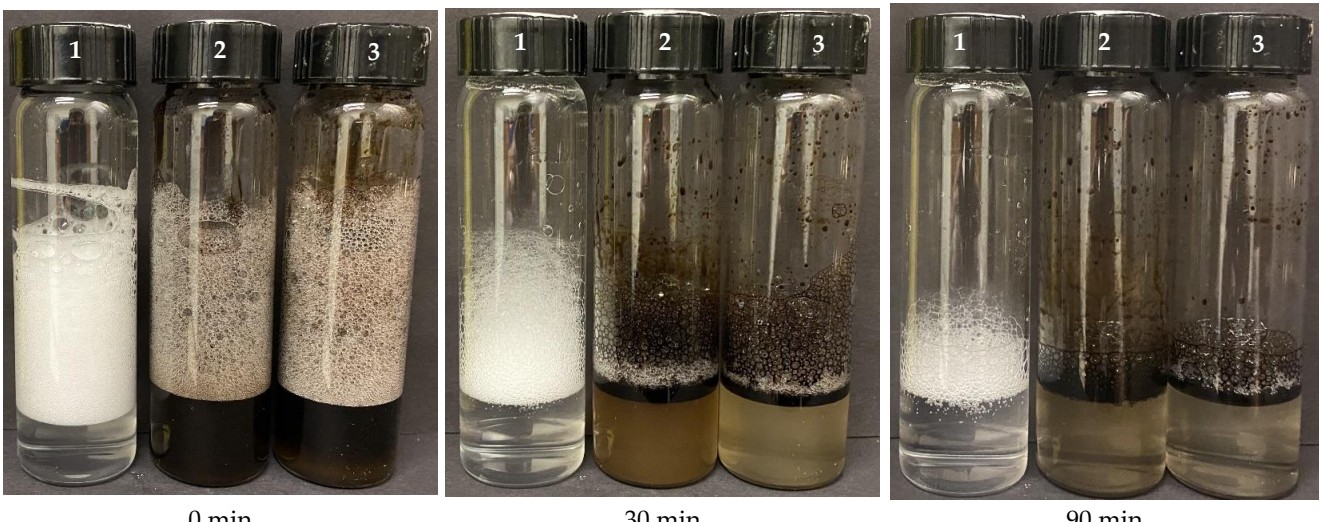

|  |  |  |
|---|---|---|
| 0 min | 30 min | 90 min |

**Figure 4.** Bulk foam stability test with APG and crude oils from Unit-A and Unit-B: (1) no oil, (2) Unit-A oil, and (3) Unit-B oil.

A plot of the foam height with time is shown in Figure 5. The initial foam height shows a minor reduction of 12% in the presence of both crude oils. For the case without oil, the foam half-life is 90 min, while for both crude oils, it is 27 min. Our extensive surfactant screening work shows that foam half-life in presence of oil being longer than 5 min often indicates a promising oil resistance during foam flow in porous media, and that poor oil tolerance causes bulk foam to collapse completely within few mins and such foam system does not yield significant mobility reduction in porous media. While it is insufficient to draw a conclusion on foam strength in porous media based solely on bulk foam tests, the ability of the APG stabilized foam to propagate in carbonate rocks in the presence of oil needs to be confirmed through coreflood experiments.

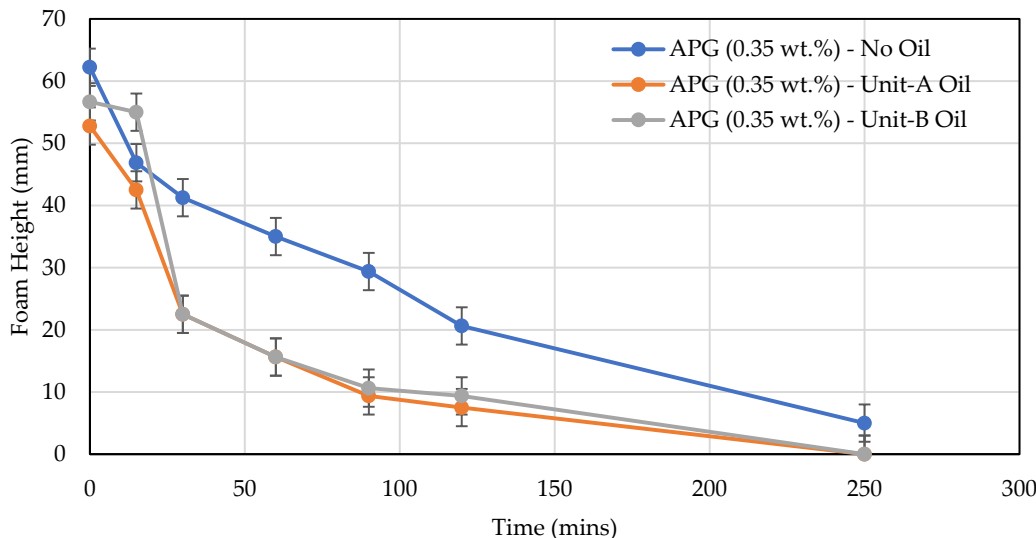

**Figure 5.** Bulk foam height (mm) as a function of time (minutes) without and with crude oils from Unit-A and Unit-B.

### 3.2. Foam Propagation in Individual Reservoir Units

Foam in porous media is commonly studied by two methods. The first method, which is used in this work, involves the simultaneous injection of surfactant solution and gas. The second method is the alternating injection of surfactant and gas slugs. As foam is treated as a separate phase made up of gas-in-water dispersions stabilized by a surfactant, the apparent viscosity of this phase at quasi-steady state may be estimated from Darcy's law as shown in Equation (5). An alternative treatment estimates the mobility reduction factor (MRF) as defined in Equation (6).

$$\mu_{apparent} = (Ak\Delta P/lu) \tag{5}$$

$$MRF = (\Delta P_{foam}/\Delta P_{Brine}) \tag{6}$$

where $\Delta p$ is the total pressure drop across the core, A is the core cross-sectional area, l the length of the core, and u is the total Darcy velocity, and k is the permeability of the core, $\Delta P_{Foam}$ is the pressure drop across the core during foam injection, and $\Delta P_{Brine}$ is the pressure drop of single-phase brine injection.

As listed in Table 3, two series of core floods were carried out under the reservoir conditions of Unit-A and Unit-B. First, Exp-1 in both units establishes the maximum possible apparent viscosity under water-wet conditions without the presence of oil. For the high permeability Unit-A, the injection rates of 2 and 5 ft/d were used, while the rate is limited to 1–3 ft/d in the low permeability Unit-B. The results are summarized in Figure 6.

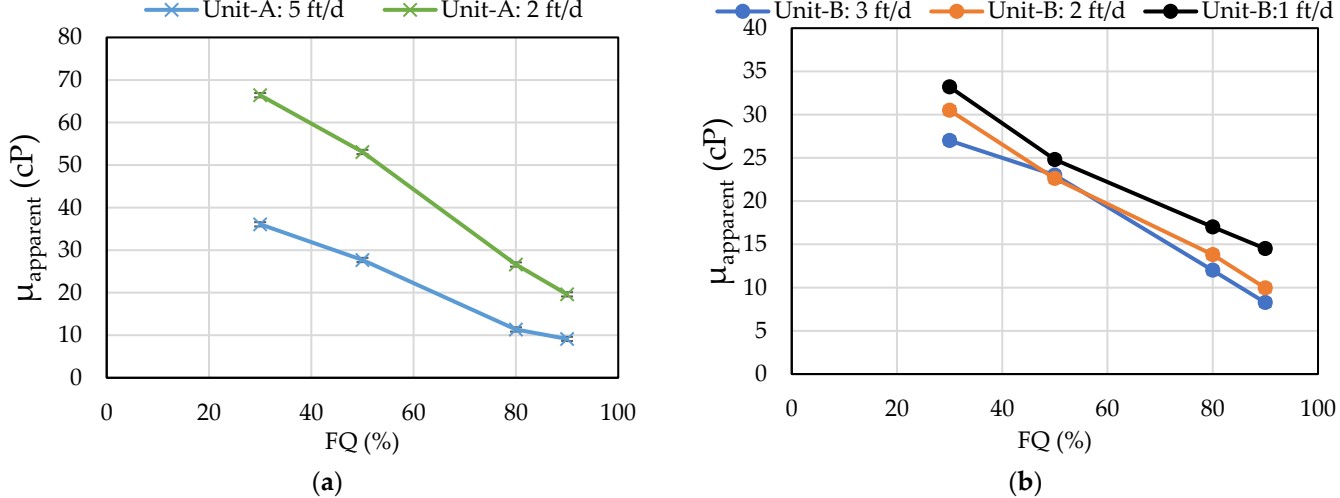

**Figure 6.** Apparent viscosity in (**a**) Unit-A Exp 1, and (**b**) Unit-B Exp 1 as a function of FQ for injection rates varying from 1–5 ft/d.

Foam in both experiments in the absence of oil has a much higher apparent viscosity at lower foam qualities (30 and 50%). As shown in Figure 6a, foam in Unit-A ranges between 10–30 cP at higher foam qualities (dry foam), and 28–65 cP at low foam qualities (wet foam). In the lower permeability Unit-B, weaker foam is generated between 6–15 cP at (80–90% *FQ*) and 22–34 cP at (30–50% *FQ*). The observed *FQ* dependence of foam viscosity is very distinct from the commonly observed steady state foam behavior that exhibits a transition *FQ* at which foam viscosity reaches a maximum, and below which foam viscosity increases with *FQ* (referred to as low foam quality regime in the literature) [18]. This regime is not observed with the APG surfactant. However, both experiments display typical shear thinning foam behavior, whereby foam viscosity is higher at lower injection rates (shear rates).

The next series of core floods used oil wet cores saturated with oil from Unit-A and Unit-B field samples. Water flooding was carried out to reduce oil saturation to residual ($S_{orw}$). Co-injection of brine and gas was conducted to simulate a WAG process for tertiary oil recovery, whereby the oil saturation further decreases to another residual value (Sorg). During the foam flood stage, no significant oil production was observed. This process was followed with foam injection at 2 ft/d and 30–50% FQ. Most of the oil production in these single-core floods occurred in the water flood and brine-gas co-injection stages and represents the best possible oil recovery for each reservoir unit.

Figure 7 summarizes the oil recovery and foam viscosity for Unit-A Exp 2 (25% porosity, 100.8 mD) and Unit-B Exp 2 (31% porosity, 19.7 mD) under oil wet conditions and residual oil saturation. For Unit-A, the waterflood stage at 2 ft/d produces 33% of the original oil in place (OIIP), while the brine-gas injection stage at 2 and 5 ft/d with both at 50% injection gas fraction further raises the recovery to 78 and 83%, successively. Foam is then flooded at Sorg ~11.4% at 2 ft/d, resulting in the foam viscosity of 23 cP for 50% FQ and 27 cP for 30% FQ. This result shows a 55% reduction in foam viscosity from the water-wet case in the absence of oil (Unit-A Exp 1).

In Unit-B Exp 2, the waterflood produces 63% of the oil, and the brine-gas injection stage raises the oil recovery to a maximum of 72%. Foam injection at Sorg ~18.9% and 2 ft/d generates foam that is 21 and 19 cP at 30 and 50% FQ, respectively, which shows an average reduction of 25% in foam viscosity from the water-wet case without oil (Unit-B Exp 1). The above foam viscosity results are consistent with the foam stability characterization (oil spreading tendency and bulk foam stability tests), which predicated that APG foam could propagate in the presence of Unit-A and Unit-B crude oils. The above oil recovery results in both experiments represents oil displacements in the absence of any conformance issues. In a realistic production scenario, injected gas will preferentially flow in the higher

permeability Unit-A, leaving Unit-B poorly gas processed. Other conformance issues such as gravity segregation will further reduce the overall oil recovery factor.

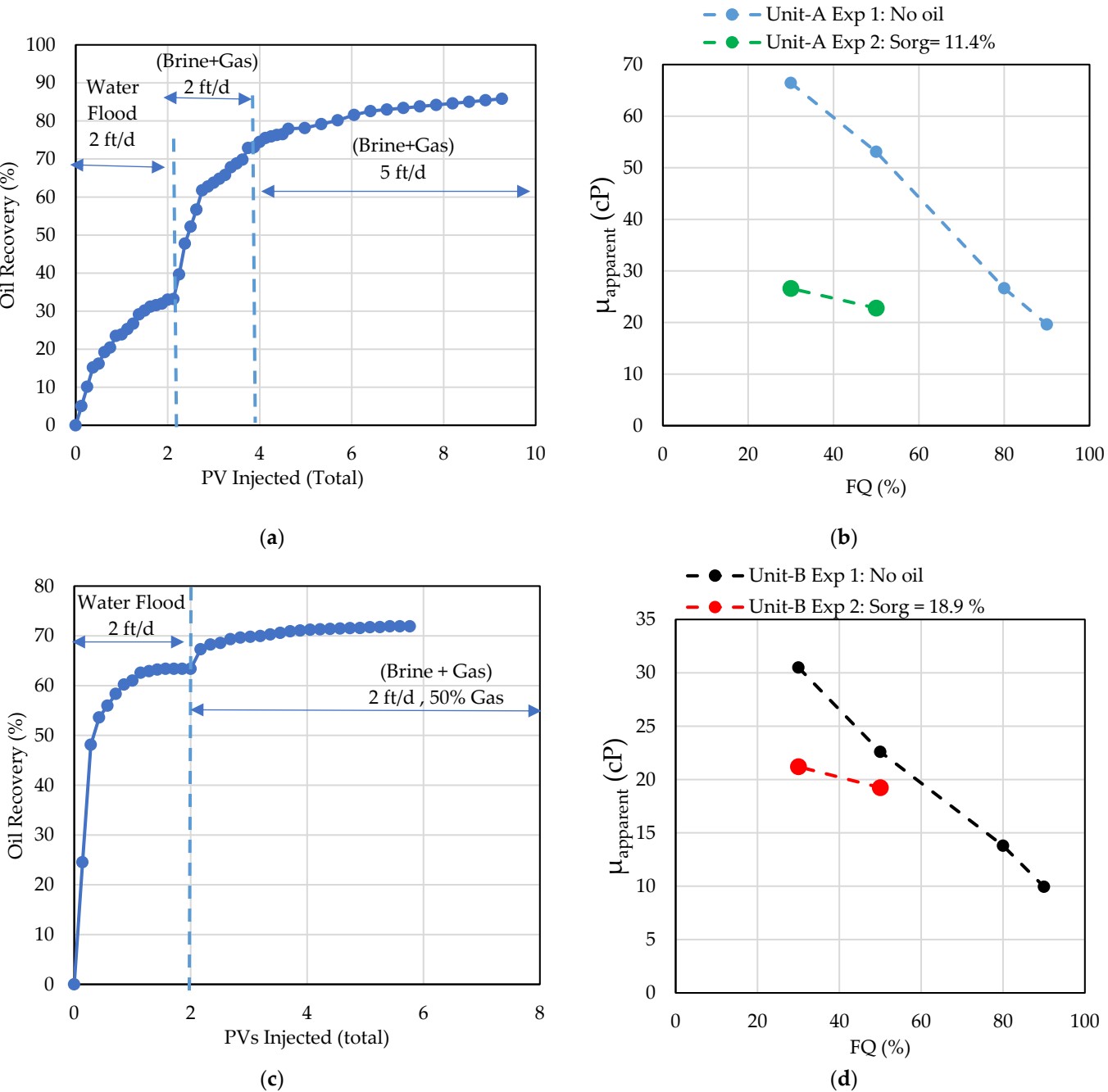

**Figure 7.** Results of Unit-A Exp 2: (**a**) oil recovery (%), and (**b**) apparent viscosity at 2 ft/d and Sorg = 11.4%. Unit-B Exp 2 results: (**c**) oil recovery (%), and (**d**) the apparent viscosity at 2 ft/d and Sorg = 18.9%. Gas-brine flooding is at 50% injection gas fraction.

### 3.3. Foam Propagation in Dual-Unit Reservoir

To fully demonstrate the viability of foam in improving immiscible gas EOR in this type of heterogeneous carbonate reservoirs, three dual-core flooding experiments were designed. In DI-Exp 1, foam propagation and gas saturation over time are related to fluid diversion between the reservoir units. In DI-Exp 2 and DI-Exp 3, the results focus on oil recovery associated with foam propagation.

### 3.3.1. Dual-Core Flood in Water-Wet Conditions without Oil

Figure 8 shows the average system pressure across both cores during the various injection stages. Since both cores share a single inlet, and are set to the same outlet pressure, the resulting pressure drop across both cores is the same. In this parallel flow system, flow rate and fluid composition variations were monitored. Figures 9 and 10 show the fractional flow, and gas saturation, respectively.

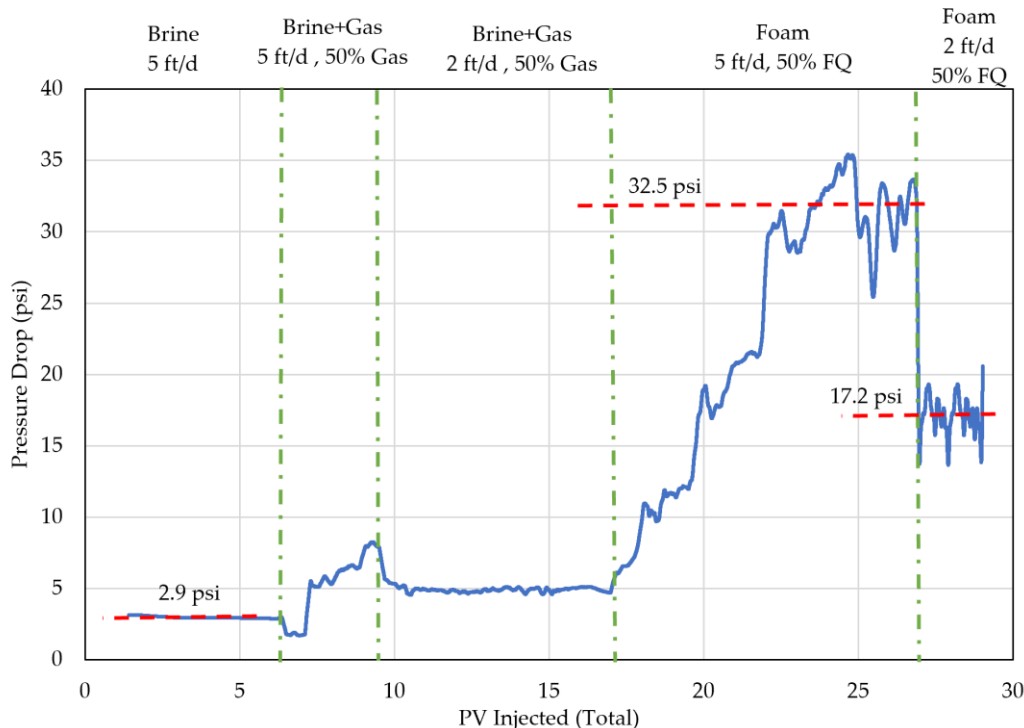

**Figure 8.** Pressure drop across both cores (Unit-A and Unit-B) for DI-Exp 1.

As shown above in Figure 8, foam injection at 5 ft/d and 50% FQ causes a rapid increase in the pressure drop up to 32.5 psi, resulting in an MRF of 11.2 over the baseline brine injection. Foam injection at 2 ft/d and 50% FQ results in a 17.2 psi pressure drop or an MRF of 14.8. As shown in Figure 9, the fractional flow during the brine and brine-gas injection is consistently 80–95% through Unit-A and 5–20% through Unit-B. However, after foam is generated, the fractional flow is modified in less than 1.5 PVs of foam injection and becomes uniform (50–50%) for 5 ft/d, and 60–40% for 2 ft/d. Furthermore, the gas saturation is seen to be less than 10% of the total PV during the brine-gas injection, and steadily rises above 15% during foam injection at the 5 ft/d (Figure 10). It is noted that an accurate determination of gas saturation was not possible beyond 20 PVs due to strong foam overflowing at both core effluent outlets, preventing an accurate assessment of liquid effluent from each core.

The low gas saturation during brine-gas injection suggests that gas can finger through the reservoir and break through early, bypassing oil during WAG injection in the field. No detectable rise in the gas saturation occurs once the gas has an established flow path through the core. However, foam propagation helps increase gas saturation in the core, as well as increase the probability of contact between the injected gas and trapped oil in small or bypassed pores as further discussed in the following section.

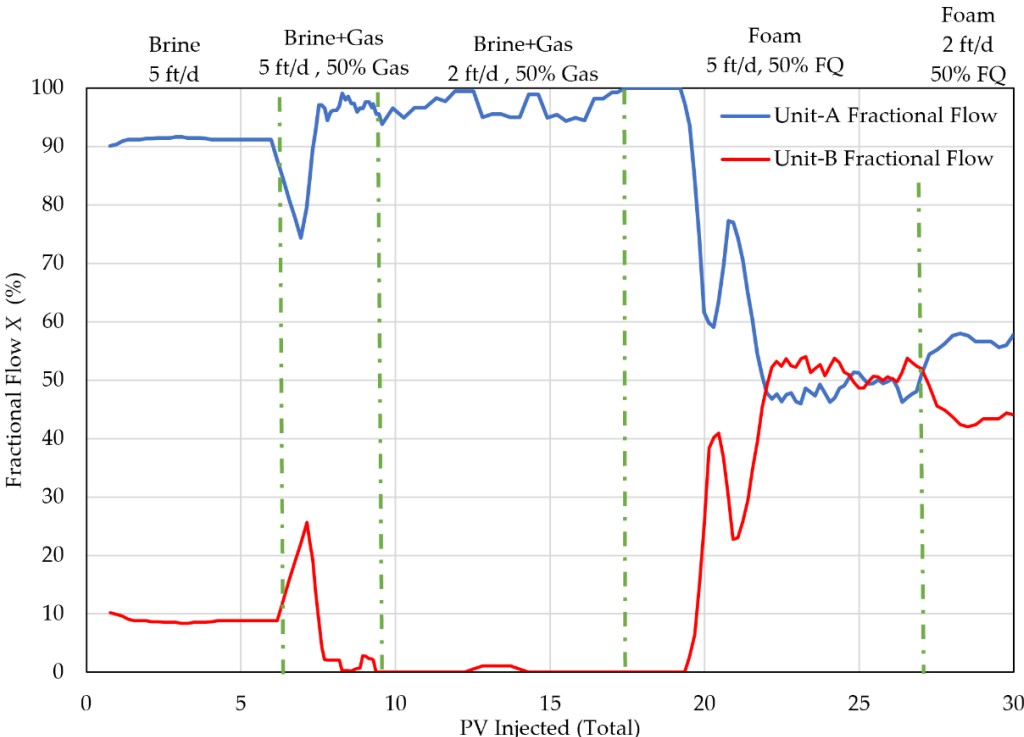

**Figure 9.** Fractional flow through Unit-A and Unit-B cores for DI-Exp 1.

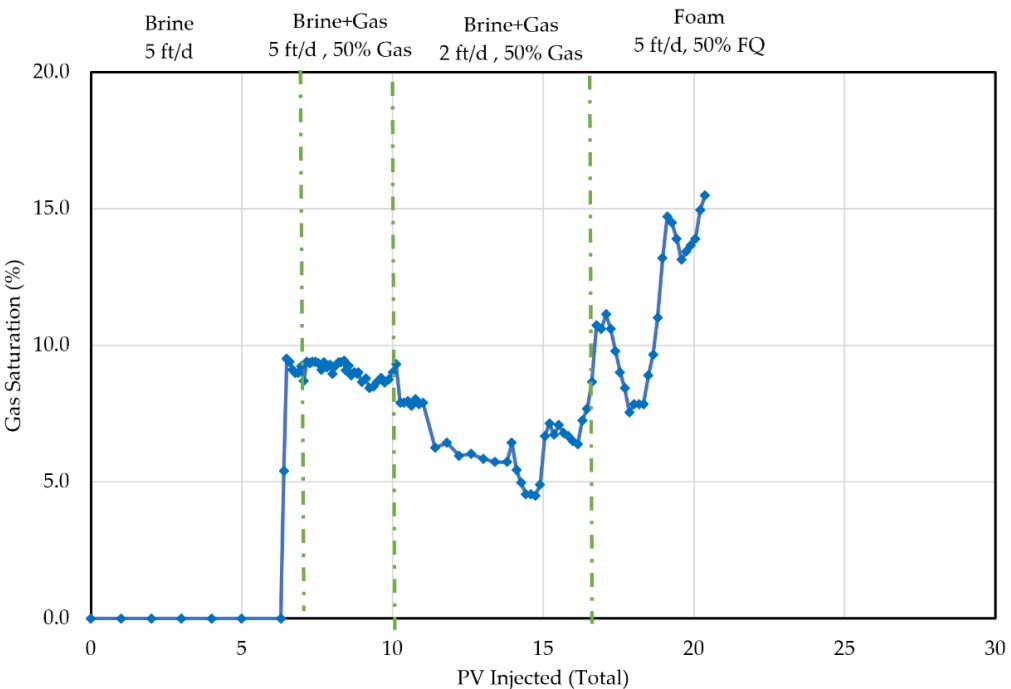

**Figure 10.** Combined gas saturation in both cores for DI-Exp 1.

### 3.3.2. Dual-Core Floods under Oil Wet Conditions

DI-Exp 2 is aimed to prove the viability of foam under the reservoir's oil wet conditions. The combined oil saturation and system pressure drop for this experiment are presented in Figure 11. A shown in this figure, the waterflood stage reaches a steady state in terms of the system pressure drop (23 psi) and combined oil saturation (24%) within 2–3 PVs. Foam injection at 5 ft/d and 50% FQ is then conducted, which results in a further reduction in oil saturation to 10%, and a pressure drop increase to 60 psi at quasi-steady state. This

represents an MRF of 2.6 over the waterflood pressure drop. Most of the additional oil production occurs within the first 3 PVs of foam injection.

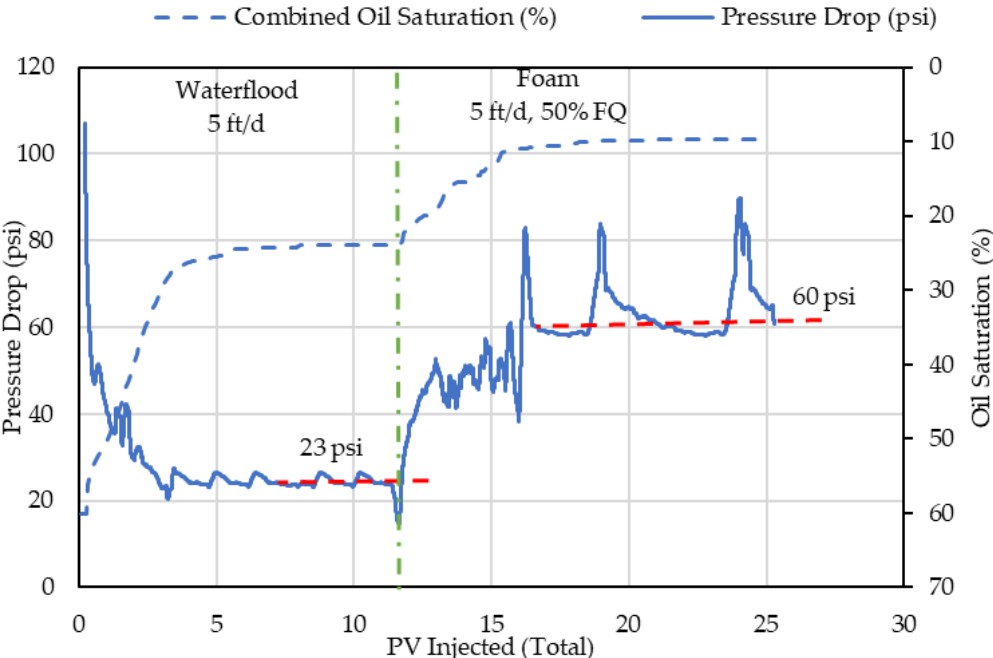

**Figure 11.** Pressure drop across both cores and the combined oil saturation for DI-Exp 2.

Fractional flow and oil recovery from each reservoir unit are shown in Figures 12 and 13, respectively. In Figure 12, fractional flow during the late time waterflood shows nearly 100% injection water penetrating Unit-A. During this stage most of the oil recovery occurs from Unit-A. As shown in Figure 13, 74% of the oil in Unit-A is recovered within 5 PVs, while only 46% of the oil in Unit-B is recovered.

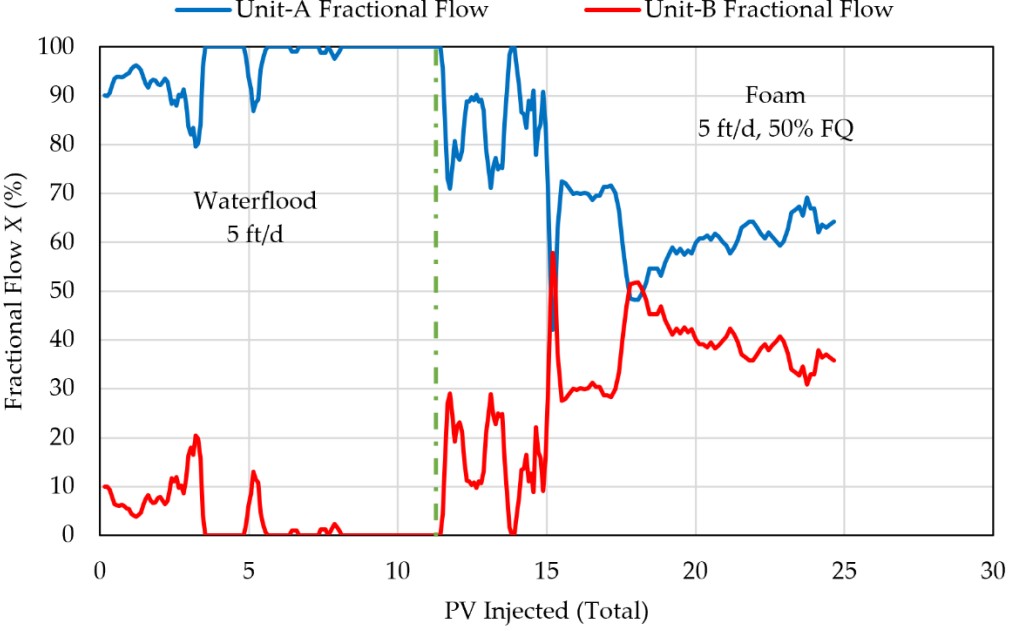

**Figure 12.** Fractional flow through Unit-A and Unit-B cores for DI-Exp 2.

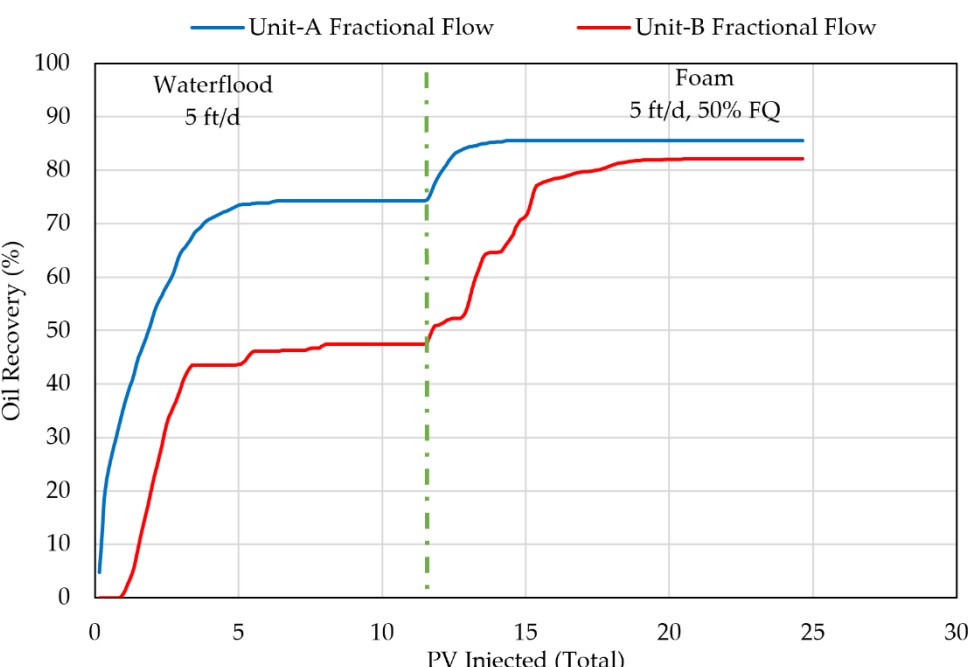

**Figure 13.** Oil recovery (%) from Unit-A and Unit-B cores for DI-Exp 2.

Fractional flow rapidly changes during the foam injection stage. First, considering the period of 11–15 PVs, the liquid fractional flow reduces from 100 to 80% through Unit-A or increases to 20% through Unit-B. This is associated with a steep rise in oil recovery from Unit-B (from 46 to 76%). This production increase is due to foam generation in Unit-A diverting more gas and water to Unit-B leading to better sweep efficiency. A sharp rise in production from 74 to 85% also occurs in Unit-A. Such an increase in production from this flow unit is directly attributed to foam induced gas mobility reduction (indicated by the pressure drop buildup shown in Figure 11). The average fractional flow continues to decrease to around 60% in Unit-A beyond 15 PVs as foam continues to develop stronger in this unit (Figure 11), which leads to a further improvement of oil recovery from Unit-B up to 82% of the OIIP.

The results from the DI-Exp 2 experiment demonstrates the dynamic nature of foam behavior and its impact on fluid distribution in macroscopically heterogeneous reservoirs. As foam builds up in the high permeability oil zone (Unit-A pressure buildup in Figure 11), gas and water are diverted into lower permeability zone (Unit-B), resulting in a remarkable increase in oil production (Figures 12 and 13). Despite the diversion of surfactant and gas into Unit-B, the system pressure drop continues to rise, indicating the robustness of the foam generation in this model layered reservoir even under oil-wet condition.

### 3.3.3. Dual-Core Floods at Elevated Oil Saturation

In DI-Exp 3, the cores were water-flooded for only about 1 PV, and then foam injection started at 5 ft/d and 50%. The system pressure drop and combined oil saturations are shown below in Figure 14. The waterflood stage reduces the combined oil saturation to 40%. The pressure drop at the end of the waterflood is 50 psi. Foam injection further reduces oil saturation to 14% as the pressure drop increases to 75 psi equivalent to an MRF of 1.5 over the end of the waterflood in this experiment. A detailed overview of the fractional flow and oil recovery from Unit-A and Unit-B cores can be seen below in Figures 15 and 16.

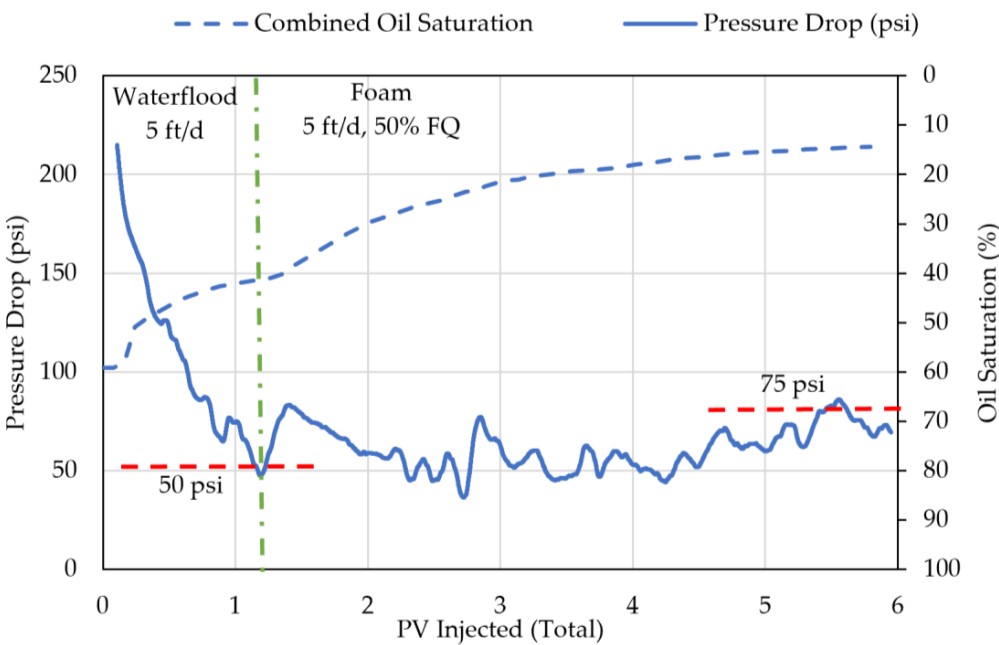

**Figure 14.** Pressure drop across both cores and the combined oil saturation for DI-Exp 3.

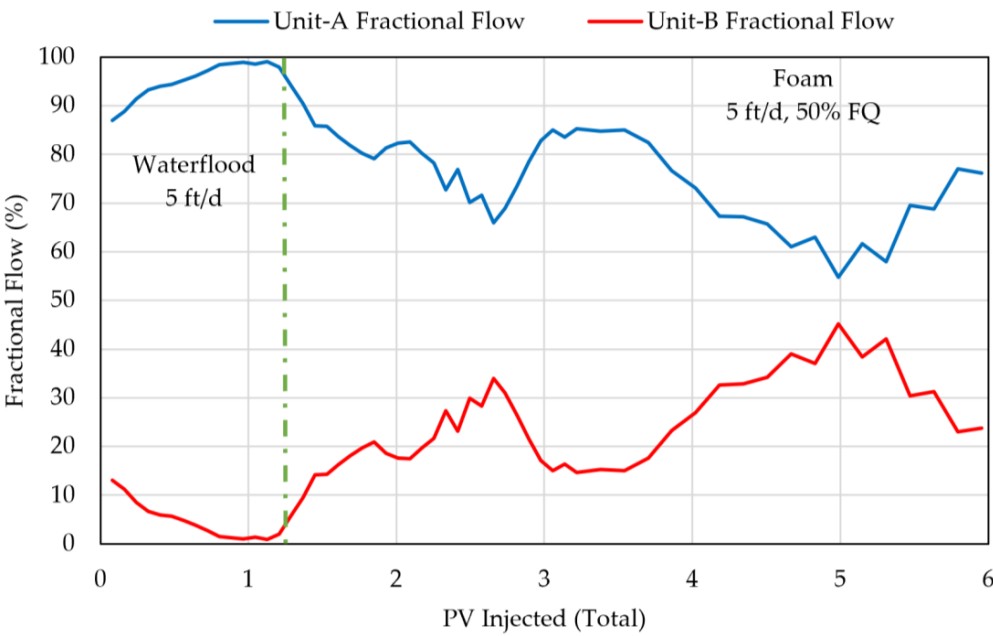

**Figure 15.** Fractional flow through Unit-A and Unit-B cores for DI-Exp 3.

Figure 15 shows nearly 100% of the liquid flowing through Unit-A during the waterflood, which is quite similar to the results from DI-Exp 2. During this stage, 44% of the oil is recovered from Unit-A, while only 17.5% is recovered from Unit-B. As foam injection propagates between 1.1–6 PVs, it leads to a gradual reduction in fractional flow through Unit-A and an increase through Unit-B. The fractional flows fluctuate, but hover between 60 and 80% through Unit-A. The oil recovery profile from both units is similar, indicating even more sweep of oil from both units simultaneously, reaching an ultimate recovery of 81% from Unit-A and 71% from Unit-B after a total of 6 injected PVs.

The system pressure drop rises in DI-Exp 3 (25 psi within 5 PVs) is clearly slower than that in DI-Exp 2 (38 psi in 3.5 PVs). However, the slower foam propagation did not negatively affect its ability to provide conformance and improve the overall sweep efficiency, as evidenced by the results shown in Figure 15.

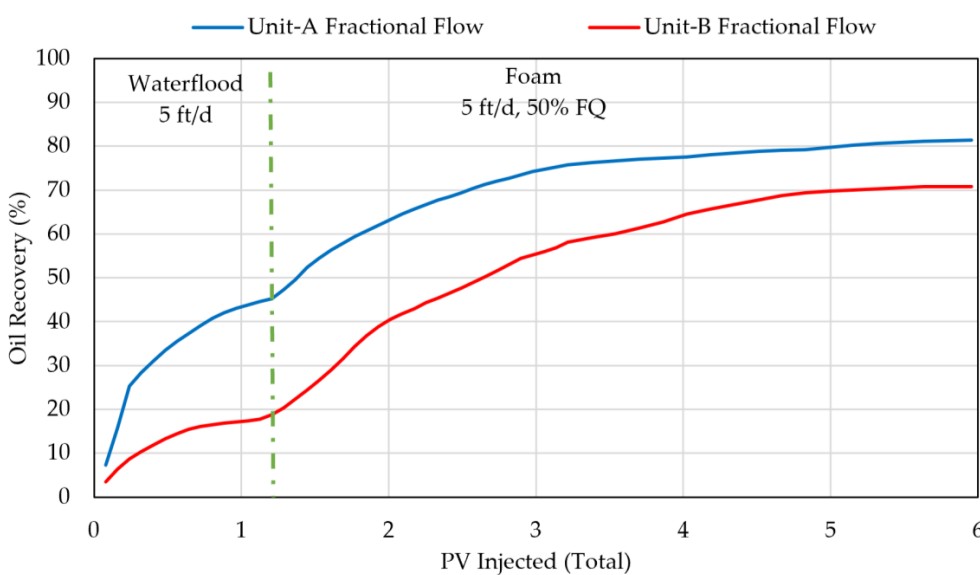

**Figure 16.** Oil recovery (%) from Unit-A and Unit-B cores for DI-Exp 3.

The higher mobile oil saturations were not detrimental to the goal of the injected surfactant and gas, which is to provide conformance control to the injected fluids. This leads to an important consideration for foam field implementation, whereby a reduced foam viscosity when in contact with high oil saturations (evidenced by slow rise in pressure, or slow reduction in injectivity) does not reduce the ability of foam to improve immiscible gas flood efficiency. Furthermore, the results of DI-Exp 3 suggest that foam-assisted immiscible gas flooding could be a viable process to replace waterflooding for enhanced oil recovery.

## 4. Conclusions

This work explored foam injection as a means of improving immiscible gas flood efficiency in a layered carbonate reservoir composed of a high permeability layer (Unit-A), and a low permeability layer (Unit-B). The selected surfactant (APG) was shown to be stable at the reservoir salinity (144,000 ppm) and temperature (50 °C). The fluid characterization results lead to the following key conclusions:

- APG forms type-I microemulsions with both crude oils. However, the presence of this microemulsion does not significantly impact bulk foam stability.
- More emulsification occurs with the crude oil of Unit-A, supported by the higher lamella number for this oil.
- Both Entering and Spreading coefficients are highly negative, indicating stable foam.

For individual reservoir units, an increase in foam viscosity with permeability was found for both water-wet and oil-wet conditions, which is a very favorable foam rheological property for fluid diversion in reservoirs with permeability variations. Moreover, it is observed for the first time that foam viscosity consistently decreases with increasing foam quality from 30 to 90%, regardless of the rock wettability. While foam viscosity was not determined for foam quality below 30%, it is expected to decrease as foam quality decreases further below 30% because foam becomes so wet that its viscosity should approach water viscosity.

Foam behaviors in the dual-core systems are consistent with those in the individual cores. Some distinct findings can be summarized as follows.

- Foam injection into water-wet cores without oil can provide uniform mobility of the injected fluids in both cores with a significant permeability contrast, supported by the foam viscosity-permeability relationship observed from the single-core flooding experiments.
- Foam could propagate in an oil-wet environment, resulting in a significant improvement of oil recovery from the low permeability zone (46 to 82% for Unit-B).

- Elevated oil saturation due to a premature waterflood does not appear to completely inhibit foam propagation as even more incremental oil production, particularly from the low permeability zone (Unit-B), could be obtained with foam. This experiment suggests that foam can be a promising conformance agent for heterogeneous carbonate reservoirs with varying oil saturation.

However, future work is still needed to investigate the effect broader permeability contrast and the scalability of dynamic foam behaviors.

**Author Contributions:** Conceptualization, M.T. and Q.N.; methodology, M.T. and Q.N.; validation, Q.N. and P.P.; writing—original draft preparation, M.T.; writing—review and editing, Q.N.; supervision, Q.N.; project administration, P.P.; funding acquisition, Q.N. All authors have read and agreed to the published version of the manuscript.

**Funding:** This research received no external funding.

**Data Availability Statement:** Not applicable.

**Conflicts of Interest:** The authors declare no conflict of interest. The funders had no role in the design of the study; in the collection, analyses, or interpretation of data; in the writing of the manuscript or in the decision to publish the results.

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
