# Peer review of "Experimental Evaluation of Foam Diversion for EOR in Heterogeneous Carbonate Rocks"

_colloids, doi:10.3390/colloids6040063_

Round 1

Reviewer 1 Report

The paper is interesting and well written. I have a few minor comments to improve the quality of the work: 

·         Cores from different outcrops were used for tests. Did it affect the dual-core injection results? Do they achieve a similar initial wettability? It is better to explain this point in the manuscript.

·         What is your conclusion from Figure 3-c about the application of this surfactant? Is there any concern of high viscosity due to the mixing at reservoir salinity?

·         How is it possible to conclude “APG can generate foam with good stability in the presence of both crude oils” from Figure 5? What is the criteria here to approve these foams? It is suggested to add it to the manuscript.

·         As FQ is an important parameter in your study, I suggest adding a few sentences to explain it more.

·         How much is the oil recovery during foam flooding (Figure 7)? The figure is confusing as it seems that only a gas/water flooding was studied.

·         Dual core flooding results are well presented. There is only a question regarding the pressure drop presented in figures such as figure 11. As there are two different cores in the system, to which porous media is the presented data related?

Reviewer 2 Report

The article “Experimental Evaluation of Foam Diversion for Gas EOR in a Heterogenous Reservoirs” was submitted to Colloids and Interfaces.

General observations:

In this manuscript, the authors investigated foam as a solution for gas conformance issues applied to heterogenous carbonate reservoirs. They first studied the interactions (emulsification, surface tension, foamability) between crude oil and brine surfactant solution at reservoir conditions. The main part of the article deals with single and dual core injection in water wet and oil wet conditions.

The english language and style are correct, only minor spelling are required.

The article is well introduced and documented as there are many references. Although the "Materials and Methods" section is well detailed, some important information is missing, such as the reference of the tensiometer. Concerning the ‘Results and Discussion’ part, the fluid characterization is more qualitative than quantitative and could have been a little more investigated. The foam propagation sections are well described and the results presented are interesting. The conclusions are well supported by the experimental results obtained.

Therefore, for all of the above reasons, I recommend the publication of this work in Colloids and Interfaces after addressing the following comments and questions.

Observations and comments:

TDS is not defined in the article.

Please add the definition of foam quality in the article.

2. Materials and Methods:

No information is given on the APG surfactant used : name ? formula ? molar mass ? HLB ? CMC ? …

Page 3 lines 106 – 115 : this part is repetitive of the end of the introduction and should instead appear in section 1.4 to avoid confusion.

Table 1 should also mention the ionic strength of the brine.

Page 3 lines 122- 126 : the densities of crude oils should also appear in term of API and the authors should specify what is the class of the crude oil at room temperature (light? medium? … ?).

Page 4 line 157 : Please indicate the supplier and the reference of the tensiometer.

Section 1.3 : The phase density of the phases is the parameter which allows to evaluate the value of the tension, however it is not mentioned here.

3. Results and discussion

Section 3.1.2 :

In the section 1.2, the authors specified the emulsion is made by ‘mixed gently’ the surfactant solution and the crude oil in the glass pipet. It is well known that emulsion stability depends on the initial size of the droplets. Can the authors comment on the fact that the emulsion produced in the glass pipet is well representative of the emulsion that can form in the porous medium?

It is not clear from Figure 3c that an emulsion intermediate phase is present.

Section 3.1.3 :

The calculation of E, S, and L in Table 4 do not match exactly for Unit A. For example, I get respectively -34.41 for E.

No standard deviation is mentioned for the IFT/surface tension measurements. How many times were the measurements repeated?  The drop tensiometer often has problems to evaluate tensions lower than 1 mN/m because the drop detaches itself. Have you encountered these problems? If so, how did you decide on the value of the interfacial tension?

Section 3.1.4 :

Is the surfactant concentration used for the foam below or above the CMC?

What is the initial foam quality generated in your bottle tests?

Page 9 lines 285-286 : ‘This indicates that APG can generate foam with good stability in the presence of both crude oils’. What is your criterion for judging the good stability of foam?

The foam generated in the porous medium is a 2D foam rather than a 3D foam, but your foamability test was in 3D, how do you relate the foamability of the 3D to the 2D?

Section 3.2 :

Why is there no error bar for apparent viscosity measurements? Has a duplicate or triplicate been done? If not, could there at least be a standard deviation related to the measurement of ΔP.

Section 3.2 to 3.3.2 :

The rest of the article focuses on the injection of the foam in the porous medium with measurement of the oil recovery factor and pressure drop.  Although the experiments in porous media are tedious and not very reproducible, it would have been good to indicate in the article that each experiment was repeated at least 1 time and to add some curves in supporting information.

Reviewer 3 Report

The paper presents an experimental work showing foam stability and injection in single and dual cores. Some terms are described later in the paper than their first appearance. Defining them sooner can improve the readability. Describing the mechanisms behind some trends will also provide more information to the readers. Specific comments are found below.

Title: Gas EOR should just be EOR and this is a lab investigation, so not a reservoir.

Abstract: What kind of foam was used? You should make it clear what kind of material you study. Some descriptions are difficult to interpret.

P2 Explain how wettability affects WAG and why oil wetness is considered negative.

P2 Explain why foam apparent viscosity increases with higher permeability.

How do you define scalability of viscosity-permeability relationship?

Table 2 What are the values representing? You must have achieved more accurate values of porosity and permeability?

1.1 Define foam halflife.

Table 3 does not show which experiment has highest viscosity. This observation should anyway be in results.

What were the initial saturations? Should be listed in table 3.

3.1.2 Explain the difference between foam and unwanted emulsions.

3.1.3 Add sources for these coefficients and their relation to stability.

Use consistent notation in eq 4. What is the foam plateau border radius?

P9 Why is 27 minutes considered good stability? What does that indicate in the reservoir?

The results are presented very clearly.

P19 ‘…observed for the first time that foam viscosity constantly decreases’ If you had included more foam quality points you should see a peak since the water viscosity would have been gained at zero quality.
